

# Development of an *in situ* Acoustic Anemometer to Measure Wind in the Stratosphere for SENSOR

Liang Song[1,2], Xiong Hu[1], Feng Wei[1], Zhaoai Yan[1,3], Qingchen Xu[1], Cui Tu[1,3]

[1]Key Laboratory of Science and Technology on Environmental Space Situation Awareness, National
Space Science Center, Chinese Academy of Sciences, Beijing 100190, China
[2]College of Earth Sciences, University of Chinese Academy of Sciences, Beijing 100049, China
[3]College of Materials Science and Opto-Electronic Technology, University of Chinese Academy of
Sciences, Beijing 100049, China

*Correspondence to*: Liang Song (songliang@nssc.ac.cn)

**Abstract**. The Stratospheric Environmental respoNses to Solar stORms (SENSOR) campaign investigates the influence of solar storms on the stratosphere. This campaign employs a long-duration zero-pressure balloon as a platform to carry multiple types of payloads during a series of flight experiments in the mid-latitude stratosphere from 2019 to 2022. This article describes the development and testing of an acoustic anemometer for obtaining *in situ* wind measurements along the balloon
trajectory. Developing this anemometer was necessary, as there is no existing commercial off-the-shelf product, to the authors' knowledge, capable of obtaining *in situ* wind measurements on a high-altitude balloon or other similar floating platform in the stratosphere. The anemometer is also equipped with temperature, pressure, and humidity sensors from a Temperature-Pressure-Humidity measurement module, inherited from a radiosonde developed for sounding balloons. The acoustic anemometer and
other sensors were used in a flight experiment of the SENSOR campaign that took place in the Da chaidan District (95.37°E, 37.74°N) on 4 September 2019. Three-dimensional wind speed observations, which were obtained during level flight at an altitude of around 25 km, are presented. A preliminary analysis of the measurements yielded by the anemometer are also discussed. In addition to wind speed measurements, temperature, pressure, and relative humidity measurements during ascent are compared to observations
from a nearby radiosonde launched four hours earlier. The problems experienced by the acoustic anemometer during the 2019 experiment show that the acoustic anemometer must be improved for future experiments in the SENSOR campaign.

## 1. Introduction

The response of the stratosphere to solar activities is an important scientific problem in the study of the
solar-terrestrial relationship. In theory, it is known that solar flares, proton events, and Coronal Mass Ejections can cause sudden and global violent disturbances in the stratosphere, and that atmospheric waves may be stimulated by short-term solar storms(Hood, 1987; Brasseur, 1993; Shindell et al., 2001; Pudovkin, 2004; Gopalswamy et al., 2006; Labitzke, 2006; Thomas et al., 2007; Gray et al., 2010; Shi et al., 2018). However, due to the lack of high-resolution, and continuous observational data in the
stratosphere, it is impossible to accurately describe how and to what extent solar activities affect the mid-latitude stratosphere. The campaign of Stratospheric Environmental respoNses to Solar stORms (SENSOR), focusing on the above scientific research problems, has been developed (Hu, 2018). SENSOR employs a long-duration zero-pressure balloon, which is drifted above an altitude of 20km, as



the main platform to carry multiple types of payloads for conducting a series of flight experiments in the mid-latitude stratosphere during the period between 2019 and 2022. These experiments take place during the ascending phase of solar activity cycle. The balloon-borne payloads can provide in-situ measurements of meteorology parameters, chemical components, electric and magnetic fields, solar UV and the neutron radiations in the stratosphere, which would be used to study their small-scale variations and the possible relationship of these variations to solar activities. An acoustic anemometer, which has been widely used

in terrestrial environmental research due to its fast response time and high sensitivity (Kato et al., 1992; Zacharias et al., 2011; Liu et al., 2016; Bogena et al., 2018; Grachev et al., 2018; Al-Jiboori and Jaber, 2019), is a clear choice as one of the payloads to provide measurements of wind for SENSOR.

To the authors' knowledge, there is no existing commercial off-the-shelf product that can obtain *in situ* wind measurements at altitudes exceeding 20 km, which are the float altitudes of the balloon used in the

SENSOR campaign. After a sonic anemometer was used in the 1950s O'Neill experiment to study turbulent fluxes in the atmospheric boundary layer (Suomi, 1957), attempts have been made to employ a similar instrument in more extreme stratospheric conditions. In the 1970s, Ovarlez et al. (1978) developed a sonic anemometer that was carried on a high-altitude balloon in 1990 to detect stratospheric fluctuations related to the Andes Mountains (de La Torre et al., 1994; de la Torre et al., 1996; Maruca et

al., 2017). A 1-D anemometer, developed by Banfield et al. (2016) for Mars, was carried on a terrestrial stratospheric balloon to verify its survivability under stratospheric atmospheric conditions that are similar to those experienced on the Martian surface. Maruca et al. (2017) employed a commercial acoustic anemometer, with only modest adjustments, on a high-altitude balloon to research turbulence and obtained measurements up to an altitude of about 18 km. Despite these exceptions, the use of acoustic

anemometers to study microscale meteorology has hitherto largely been limited to the troposphere (Siebert et al., 2003; Tjernström et al., 2004; Barthelmie et al., 2014; Canut et al., 2016; Maruca et al., 2017; Bodini et al., 2018; Egerer et al., 2019).

The major challenges for applying an acoustic anemometer in the stratosphere, where has significant difference from terrestrial environment, are the low temperatures with extremes approaching ~-70°C and

the low pressures of 30hPa at the height where the high-altitude balloon we used is floating. These extreme conditions make acoustic signals experience seriously attenuation (detail description will be presented in section 2.2.1), which is why no shelf products can be used immediately for SENSOR campaign. In order to measure small-scale atmospheric fluctuations (period ≤1 minute), a key scientific objective of the SENSOR campaign, and be able to operate at balloon drifting altitudes, an acoustic

anemometer should be developed and must achieve the following requirements.

Table 1 The requirements for the developed anemometer

| response time | ≤1s |
| --- | --- |
| pressure | 1000hPa~30hPa |
| temperature | 30°C ~-70°C |

In this article, we focus on the development of the acoustic anemometer and the preliminary experiment performed in 2019. This article is arranged as follows. The principle of operation of acoustic anemometers and the development of this acoustic anemometer are introduced in Section 2. A detailed

description of the 2019 balloon-borne experiments and preliminary analyses of the measurements are presented in Section 3. Conclusions regarding the acoustic anemometer are described in Section 4.



## 2. Acoustic Anemometer

### 2.1. Principle of operation

An acoustic anemometer is used to measure wind speed by sensing the difference in propagation time of
the sonic signal in the windward and leeward directions caused by the movement of airflow (Coppin and
Taylor, 1983; Alberigi Quaranta et al., 1985; Fernandes et al., 2017). Taking the measurement of one-
dimensional wind velocity, for example, there is a pair of transducers that are facing to each other with
a distance of $L$ (as shown in Fig. 1). Each transducer can function as a transmitter as well as a receiver.
To measure wind speed, each transducer transmits acoustic signals, and the opposite transducer is used
as a detector to receive the signals. Due to the airflow, the flight time of sound waves in opposite
directions between the pair of transducers, denoted as $t_{ww}$ and $t_{lw}$, differs. The values of $t_{ww}$ and
$t_{lw}$ have the following relationships with the wind velocity in the along-transducer direction (denoted
as $v$):

$$t_{ww} = \frac{L}{C-v},$$                                                              (1)

$$t_{lw} = \frac{L}{C+v},$$                                                       (2)

Here, $t_{ww}$ represents the travel time of signals in the windward (against the wind) direction, while $t_{lw}$
represents the travel time in the leeward (with the wind) direction. $C$ is the speed of sound. Because $t_{ww}$
and $t_{lw}$ can be directly measured, $v$ can be derived as follows:

$$v = \frac{L}{2}\left(\frac{1}{t_{lw}} - \frac{1}{t_{ww}}\right),$$                       (3)

It should be mentioned that when the anemometer is on a high-altitude balloon, it measures the wind
speed relative to the motion of the gondola rather than the earth-relative (absolute) wind speed (Canut et
al., 2016). Thus, the absolute wind speed, denoted as $\vec{v_r}$, is vector sum of the speed obtained by the
anemometer ($\vec{v}$) and the speed of the gondola's motion ($\vec{v_G}$), measured by the Global Navigation Satellite
System (GNSS) installed on the gondola.

$$\vec{v_r} = \vec{v} + \vec{v_G},$$                                                (4)

If the relative wind is measured in the same direction as the gondola's motion, then $v_r$ can be expressed
as:

$$v_r = v + v_G,$$                                                                       (5)

Here, $v_r$, $v$ and $v_G$ represent the absolute value of the corresponding speed.

### 2.2. Instrument Design


2.2.1 Challenges

As mentioned in the introduction, what make the terrestrial sonic anemometer cannot work under
stratospheric atmosphere are the low pressures and low temperatures. Essentially, the acoustic signals
will experience severe attenuation in the extreme environment, which in turn leads to a much lower
signal-to-noise ratio (SNR) of received signals. In order to make sure our anemometer can operate at
such an altitude, the characteristics of acoustic signals propagation attenuation in the atmosphere had
been analysed.

According to Bass et al., 1990, Bass et al., 1995 and Sutherland and Bass, 2004, when the sound wave
propagates in the atmosphere, the signal attenuation caused by atmospheric absorption is mainly related



to the acoustic frequency and atmospheric pressure. The attenuation coefficients $\alpha$ in dB per meters
(dB/m) can be expressed as follows:

$$\alpha = 8.686 f^2 \left\{ 1.84 \times 10^{-11} \left(\frac{p}{p_0}\right)^{-1} \left(\frac{T}{T_0}\right)^{\frac{1}{2}} \right.$$

$$\left. + \left(\frac{T}{T_0}\right)^{-\frac{5}{2}} \times \left[ 0.01278 \frac{e^{-\frac{2239.1}{T}}}{f_{r,o} + \frac{f^2}{f_{r,o}}} + 0.1068 \frac{e^{-\frac{3352}{T}}}{f_{r,N} + \frac{f^2}{f_{r,N}}} \right] \right\}, \qquad (6)$$

Where $f$ is the acoustic frequency in Hz, $p$ is the atmospheric pressure in Pa, $p_0$ is the reference
atmospheric pressure in Pa, $T$ is the atmospheric temperature in K, $T_0 = 293.15 K$, is the reference
atmospheric temperature, $f_{r,o}$, $f_{r,N}$ are the relaxation frequency of molecular oxygen and the relaxation
frequency of molecular nitrogen, respectively:

$$f_{r,o} = \frac{p}{p_0}\left(24 + 4.04 \times 10^4 h \frac{0.02+h}{0.391+h}\right), \qquad (7)$$

$$f_{r,N} = \frac{p}{p_0}\left(\frac{T_0}{T}\right)^{1/2} \left(9 + 280h\, exp\{-4.17\left[\frac{T_0}{T}^{1/3} - 1\right]\}\right), \qquad (8)$$

Where $h$ is the molar concentration of water vapor in percent.

According to the above formulas, Fig. 2 shows the attenuation of acoustic signals at different frequencies
caused by atmospheric absorption at 0.2 m away from the sound source with height.

The atmospheric absorption attenuation of acoustic signal increases with the increase of acoustic
frequency and with the decrease of atmospheric pressure that goes down exponentially with height. At
the balloon drifting altitude of about 25km, the received signal intensity with frequency of 40kHz is at
least 10dB higher than that of signals with frequencies of above 100kHz. Therefore, to achieve higher
received SNR, the sensors with resonant frequency of 40kHz had been used in our acoustic anemometer,
which is the primary difference between the anemometer that we developed and the anemometers used
in previous high-altitude balloon studies referenced in the Introduction. Moreover, an Automatic Gain
Control (AGC) circuit is also used, different from terrestrial anemometers, to adjust its gain levels with
altitude range to obtain better SNR.

### 2.2.2 Detailed Design

The acoustic anemometer is mainly comprised of two parts: the sensors mounted outside the balloon
gondola through an aluminium alloy boom, and the electronics box installed inside the gondola. Each
part (as shown in figure 3) is described in detail below.

The acoustic anemometer employs three pairs of ultrasonic transducers arranged in a three-dimensional
structure to measure wind. As mentioned above, the balloon flies in an environment of low temperature
and low pressure, causing the acoustic signals to experience more attenuation than they would in the
lower troposphere. To the authors' knowledge, most commercial acoustic anemometers operate at
frequencies above 100 kHz, which would result in severe attenuation of acoustic signals under the
conditions experienced in the stratosphere during SENSOR, possibly even making the signals
indistinguishable from background noise. To improve the signal-to-noise ratio (SNR) as much as possible,
we have chosen ultrasonic transducers that operate at a lower frequency of 40 kHz. In order to minimize
the influence of transducer shadowing effects, the distance between sensors should be as far as possible,
but this will reduce the signal-to-noise ratio of received signals, or even fail to receive signals at level
flight altitude. Therefore, the choice of a distance of 0.2m between transducers is a compromise. The





received signal is amplified immediately by an ultra-low noise preamplifier to further improve the SNR, and an Automatic Gain Control (AGC) circuit is also used, to adjust its gain levels by altitude range, because the received signal decreases as altitude increases. The different gain levels are determined by ground testing in a vacuum chamber.

The transducers are installed on a bracket with ring structures, which are manufactured by 3-D printing to ensure that each transducer is aligned with the opposite one to maximize the SNR and to ensure that the distance between transducers are precise. The preamplifier and AGC circuits are located at the bottom of the bracket instead of the electronic box to avoid transmission loss of the signal caused by the long cable between transducers and the electronic box.

During flight, temperatures outside of the gondola can drop to as low as −70°C, with the lowest temperatures occurring when the balloon passes through the tropopause. Therefore, the transducers and electric devices used in the preamplifier and AGC circuits were chosen for a wide temperature range and were tested in a thermal vacuum chamber at temperatures as low as −70°C to ensure that the transducers and circuits function under such an environment. This extreme environment has lower temperatures than the stratosphere in which the anemometer is used in SENSOR. To further protect against extreme conditions, the spaces where the circuits are placed were also been insulated.

The amplified received signals, connected to the electronic box inside of the gondola by long cables, are sampled at 1 MHz frequency by the Analog-to-Digital Converter (ADC) on a controller unit, which is one of the three boards in the electronic box. The controller board serves as a "brain" in the electrical system of the anemometer. Its core is an onboard FPGA, which operates the normal workflow of the anemometer. It generates a pulse train for the transmitting transducer, which is outputted to a Digital-to-Analog Converter (DAC), and then amplified to about 90 V (peak to peak) by the relevant driver circuits, the second board in the electronic box. The controller board also adjusts the gain levels of the AGC circuits through another DAC according to the gondola's current altitude. Finally, the controller board employs the communication interface, RS422, with gondola's storage system; however, due to limited bandwidth, only the health data used to monitor the anemometer and a portion of the observation data can be delivered to the gondola's storage system. As the anemometer is recovered after each flight, we store the observation data sampled by the ADC on a large-capacity storage card. This card also contains other datasets, such as the command data received by the RS422 interface from gondola's flight computer, which includes GPS time, altitude, longitude, latitude, gondola attitude, and gondola speed data. According to the current sampling rate of ADC and the writing speed to SD card, the signal received by each transducer can be sampled up to 50 times per second theoretically, which means that the update rate of three-dimensional wind speed measurements can reach 50Hz. While in the current application, to reduce the burden of writing large amount of data to storage card using FPGA and hence improve the reliability and stability of the device, the signal received by each sensor sampled at 10 times per second, that is, the update rate of wind speed measurements is 10Hz, which has already met our measurement requirements.

The third board is the power supply, which converts unregulated +28 V power provided by the gondola to regulated +12 V and +8 V power for use by the other circuits. A fuse in parallel with another fuse and a power resister are added to the input to protect the +28V power supply in case the input current exceeds 5A. The DC/DC converter is protected by a surge protection circuit that limits the inrush current and the start-up voltage slew. An additional electromagnetic interference (EMI) filter is also used on the power lines.





In addition to wind speed, the anemometer incorporates sensors to measure temperature, pressure, and humidity, which are located on the bracket outside of the gondola. The measurements are delivered to the controller board through an RS232 interface on a Temperature-Pressure-Humidity measurement module (TPH module), which was inherited from a radiosonde that we developed and used in a sounding

balloon. We retained the same circuits and sensors here. The temperature, pressure, and relative humidity data were also stored on the large-capacity storage card.

To avoid the flow distortion from the gondola, a long boom with the length of about 1.8m and an elevation angle of 45° had been used to keep the sensor bracket at a distance from the gondola. As a result, according to Lenoir et al., 2011, the perturbation from the gondola has little influence on the

measurements.

### 2.3. Data Processing

In the current experimental design, we store the raw sampled data and perform post-processing when the anemometer is recovered, rather than conducting real-time online processing. To obtain wind speed following the measurement method described in Section 2.1, the acoustic signal propagation time

between transducers should be determined first. However, due to electronic delay, the acoustic signal propagation time is not measured directly as the data is obtained. Instead, the ultrasonic waves received by each transducer when there is no wind are obtained first and are stored in the storage card as the reference signals. When the anemometer experiences wind, there are time differences between the received signals and the reference signals, allowing calculation of wind speed without knowledge of the

exact electronic delay. The mathematical model employed can be illustrated as follows.

When the wind speed is zero, the measured travel time of ultrasonic waves between one pair of transducers can be inferred as

$$t_{0r} = \frac{L}{C_0} + t_h = t_0 + t_h, \tag{9}$$

where $C_0$ is the speed of sound in the current environment, $t_h$ is the electronic delay along the

transmitted signal path, and $t_0$ is the time of flight (TOF) without electronic delay.

When the transducers are subjected to wind, the measured signal travel time in the leeward direction is

$$t_{1r} = \frac{L}{C_1 + v} + t_h = t_1 + t_h, \tag{10}$$

where $C_1$ is the speed of sound in the current environment, $v$ is the wind speed along the pair of transducers, and $t_1$ is the TOF without electronic delay in the leeward direction. The time difference

between the received signal and the reference signal can be obtained by adopting cross correlation between the received ultrasonic wave and the corresponding reference wave:

$$\Delta t_1 = t_{1r} - t_{0r} = t_1 - t_0, \tag{11}$$

Thus, $t_1$ would be:

$$t_1 = t_0 + \Delta t_1 = \frac{L}{C_0} + \Delta t_1, \tag{12}$$

Therefore, from Eqs. (9-12) TOF can be obtained without knowing the exact electronic delay. Similarly, $t_2$, which is the TOF in the windward direction, can be obtained from:

$$t_2 = \frac{L}{C_0} + \Delta t_2, \tag{13}$$

The wind speed can then be obtained by substituting Eqs. (12) and (13) into Eq. (3).


The speed and attitude (include the angular rates and attitude angles) of the gondola are continuously monitored by the GNSS and Inertial Navigation Sensor (INS) installed on the gondola. With these data and the angle relationship between the anemometer and the gondola, the zonal wind, meridional wind and vertical wind can be obtained by projecting to three directions with Eq. (4).

Besides, the internal update rate of measurements is 10Hz. In order to further improve the SNR, the original sampled signals within 1s are accumulated before correlation with the reference signal in data

processing, so the default update rate of the measurements we given is 1Hz.

### 2.4. Ground experiment

To verify the functionality and data reliability of our anemometer, we conducted a comparison experiment using our anemometer and a commercial acoustic anemometer (WS500-UMB, Lufft Inc., Germany) at the summit of the Wuling Mountain (117.49°E, 40.60°N) in Xinglong County, Hebei

Province, China. Table 2 shows the basic specifications of the two devices.

Table 2 The basic specifications of the developed anemometer and WS500-UMB

|  | Developed anemometer | WS500-UMB |
|---|---|---|
| Internal sampling frequency | 10Hz | 15Hz |
| Update rate of data | 10Hz/1Hz (default) | 1Hz /0.1Hz (default) |
| Resolution | 0.04m s$^{-1}$ | 0.1m s$^{-1}$ |

The two anemometers were mounted on tripods and placed on the roof of a building at the top of the mountain. This location has an open view; obstructions are much lower than the tripods' height and at a substantial distance from the tripods (Fig. 4). Thus, the natural airflow was little-disturbed and the two

anemometers were approximately in the same airflow field. To compare the anemometers' measurements with greater temporal precision, GPS was used to synchronize the time measurements of the two anemometers.

The measurement repetition rate of the commercial anemometer was 0.1 Hz while that of our own anemometer was 1 Hz; thus, the minimum time interval for data comparison was 10 s. To reduce the

error between measurements caused by the different sampling times as much as possible, data from each anemometer were averaged in 1-minute intervals before comparison.

According to the measurements from both anemometers shown in Fig. 5a, the wind speed measured by our anemometer agrees well with the commercial anemometer's results. The wind speed at the top of Wuling Mountain ranges from 1 m s$^{-1}$ to 3 m s$^{-1}$ during the observation period with an average of

approximately 1.8 m s$^{-1}$. Wind speed differences, which are calculated by subtracting our anemometer's measurements from the commercial anemometer's measurements, reach a maximum absolute value of 0.9 m s$^{-1}$ during the evaluation period (Fig. 5b). The mean wind speed difference is −0.02 m s$^{-1}$,with a standard deviation of 0.30m s$^{-1}$. Overall, this test shows that our anemometer may have a slight high bias relative to the commercial anemometer. However, the differences between the two anemometers may be

due to different conditions experienced by the two instruments, as there is no guarantee that two anemometers measure the same microscale air mass, no matter their proximity. Overall, the similarity of the results between our anemometer and the commercial anemometer allows us to conclude that our anemometer is capable of accurately measuring wind speed and that it generates output that is reasonable.



### 3. Results and Discussions

The flight experiments of the SENSOR campaign during 2019 took place in the Da chaidan District (95.37°E, 37.74°N), Qinghai Province, China. Fig. 6 shows the photograph of the gondola with all payloads assembled before launching at the experiment site.

The balloon was launched at approximately 16:18 UTC on 4 September. The flight trajectory is shown in Fig. 7. The balloon reached its maximum altitude of approximately 25 km at 17:56 UTC; level flight

lasted for approximately 1.5 hours until 19:42 UTC when the gondola was separated from the balloon by a cutter. The anemometer was powered on at the start of the experiment and functioned until the end of the float flight. When the gondola landed, the anemometer was recovered successfully; wind, temperature, pressure, and relative humidity measurements were obtained from the flight.

In this experiment, to acquire the best SNR and avoid damaging the transducers, we had set the driver

voltages as high as possible and adjusted the AGC circuit's gain levels automatically as the balloon rose. Although everything worked normally in the ground environment, during the balloon experiment the output signals from the ultrasonic transducers unexpectedly overflowed the ADC's input voltage range after launching. The signals did not return to normal until the balloon rose above an altitude of 10 km. Additionally, large spikes in the wind speed measurements were observed, the same phenomenon as

Maruca et al. (2017) reported. To the best of the authors' knowledge, these spikes were caused by the large attenuation of acoustic signals under low pressure, which led to the misjudgement of propagation time between transducers. However, it was straightforward to eliminate these spurious wind speed spikes, as they were clearly distinct from the data immediately preceding and following them.

Figs. 8 and 9 show a 1900-s period of zonal and meridional wind speed measurements starting at 17:33:07

UTC, during which the balloon flew from an altitude of 21.3km to the level flight altitude of about 25km. The top panels of these figures show the relative zonal ($v_U$) and meridional wind ($v_V$) measured by the anemometer at update rates of 10Hz and 1Hz, respectively. The middle panels of Figs. 8 and 9 compare the wind speed measurements (blue line) with the movement speed (red line) of the gondola drifting with the background wind, where the wind speed measurements are vector sum of relative wind velocity and

the gondola speed. From these figures, we can see that the absolute values of the relative zonal and meridional wind to the gondola never exceeded 2 m s$^{-1}$ and the measurements from anemometer were more sensitive to changes in wind than the gondola movement speed was. The principal reason is that the gondola had a large volume and a mass exceeding 400 kg, thus, it always moved with the mean air flow but its zonal and meridional speeds did not change immediately from disturbances in the wind. Fig.

10 demonstrates the vertical wind speed ($v_Z$) during this period, which was almost within ±1m s$^{-1}$ and was predominately downward.

It is worthwhile pointing out that the measurements on a drifting balloon are quasi-Lagrangian type in nature. For an ideally Lagrangian observation, the balloon motion always follows air parcels on an isentropic plane. In reality, however, the balloon oscillates about several hundred meters at float level

due to the balloon's neutral buoyancy oscillation, gravity waves, turbulence, and also the ballast system that is used to regulate the fly altitude of the balloon. Thus, the balloon does not fly on an isentropic plane, but on a constant density surface (Alexander, 2003), which is considered to be a quasi-Lagrangian type motion. According to Kolmogorov's theory (Kolmogorov, 1941), the Eulerian three-dimensional local isotropic turbulence spectrum in the inertial subrange follows a $n^{-5/3}$ slope, while

the Lagrangian spectrum, as explained by Corssin (1963), exhibits a $n^{-2}$ power law in the inertial subrange, where n is frequency. Here, a preliminary analysis of the measurements during float flight from 1500s to 1900s shown in figures 8, 9 and 10 was carried out to estimate the variation of spectral





density with frequency.

Fig. 11 shows the power spectral densities of the 10Hz measurements of relative horizontal velocity ($v_h = \sqrt{v_U{}^2 + v_V{}^2}$) and vertical wind velocity during the float flight, which are evaluated using periodogram method. The orange line and magenta line represent the expected slope of the spectrum in the Euler frame of reference and the Lagrangian frame of reference, respectively. It is difficult to say which of these two slopes is closest to the measured spectra both in relative horizontal velocity and vertical wind velocity. The difference between the slope of the measured spectra and that of the ideal Lagrangian spectrum may be due to the fact that the balloon is a quasi-Lagrangian tracer of air parcels. Moreover, the extremely similar slopes of the observed spectra of relative horizontal velocity and vertical wind velocity indicate that there was isotropic turbulence exist at the floating altitude.

The results from this experiment and the preliminary spectral analysis demonstrate that the acoustic anemometer employed in this study can sense rapid changes in wind and is useful for researching small-scale wind fluctuations in the stratosphere. Though the instrument experienced problems during this experiment, the results presented here provide confidence in using this technology to measure wind in the stratosphere.

The temperature, pressure, and relative humidity experienced by the gondola were also measured with a repetition rate of 1 Hz by their respective sensors on the anemometer. These data were compared with measurements from a radiosonde launched at 12:00 UTC, approximately four hours before the experiment, from Golmud Observation Station (94.90°E, 36.42°N), as shown in Fig. 12. The radiosonde data were downloaded from the University of Wyoming website. Although the vertical resolution of the radiosonde data was much coarser than our measurements, they still could be used to verify our measurements. Comparing the results shows that the temperature and pressure measurements had good consistency, while relative humidity had the same trend in our measurements and the radiosonde measurements. The main reason for relative humidity differences may be that the two experiments were conducted four hours apart in different locations. It is also possible that one or both humidity sensors experienced systematic errors. Besides, from plot (12a), it is clear that the tropopause height over the Qinghai Tibet Plateau was about 17 km during both sets of measurements, and tropopause temperatures reached values below −70°C.

## 4. Conclusions

An acoustic anemometer has been successfully developed for the SENSOR campaign, which is carried aboard a high-altitude balloon to measure wind velocity relative to the gondola. Hence, the absolute (earth-relative) wind speed can be obtained by vector sum of the measurements from the anemometer and the gondola speed. The latter, approximately following the mean airflow, can be acquired by GNSS equipped on the gondola. This acoustic anemometer obtains wind speed measurements using the principle that the propagation time of ultrasonic signals differs between the leeward and windward directions. Additionally, the anemometer contains a radiosonde-based module that obtains temperature, pressure, and relative humidity measurements during flight.

The anemometer participated in a flight experiment of the SENSOR campaign in 2019, which took place in the Da chaidan District (95.37°E, 37.74°N). During this experiment, the anemometer obtained continuous wind velocity data at the floating altitudes of 24-25km. These measurements demonstrated the details of the small and rapid fluctuations of stratospheric winds compared to the velocity of the gondola moving with the background wind. A preliminary spectral analysis of the quasi Lagrangian type





observations during float flight was carried out. The slope of the spectrum was very close to that of the theoretical Lagrangian turbulence spectrum, indicating that the measurements of the anemometer are feasible from the perspective of scientific analysis. Results from our experiment provide confidence in using this technology to measure wind in the stratosphere, although much work remains necessary to improve the performance of the instrument. Temperature, pressure, and relative humidity data are also

obtained from the time of launch to the end of level flight. Observations from the ascent phase were compared with the radiosonde data from the Golmud Observation station (94.90°E, 36.42°N) approximately four hours earlier, and the results show very good agreement.

The measurements can be further analysed to investigate stratospheric turbulence parameters and their relation to gravity waves and atmospheric background conditions, and can also be used to estimate

momentum fluxes directly in the stratosphere from a Lagrangian viewpoint.

*Data availability*. This article focuses on the development of acoustic anemometer. Observation data from the flight experiment are not currently publicly available since the further analysis of them is under way.


*Author Contributions.* Conceptualization and methodology, SL and HX; Writing—original draft preparation, SL; Writing—review and editing, SL, HX, WF, YZ, XQ and TC; All authors have read and agreed to the published version of the manuscript.

*Competing interests.* The authors declare that they have no conflict of interest.

*Acknowledgments.* This work is supported by the Strategic Priority Research Program of the Chinese Academy of Sciences (Grant Nos. XDA17010303, XDA17010302, XDA17010301). We gratefully acknowledge Aerospace Information Research Institute, Chinese Academy of Sciences, for providing

and successfully launching the high altitude balloon. We would like to express our gratitude to University of Wyoming website for providing the radiosonde data for comparison. We thank LetPub (www.letpub.com) for its linguistic assistance and scientific consultation during the preparation of this manuscript. We warmly acknowledge the anonymous reviewers for their careful work and thoughtful comments that have helped improve this paper substantially.

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



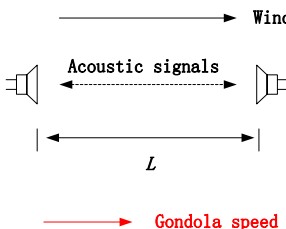

Figure 1 Diagram of the principle of measuring wind speed in a single direction using one pair of
transducers.



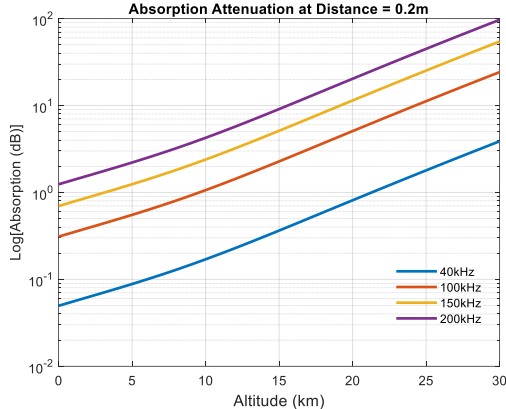

Figure 2 Atmospheric absorption attenuation of different frequencies of acoustic signals.




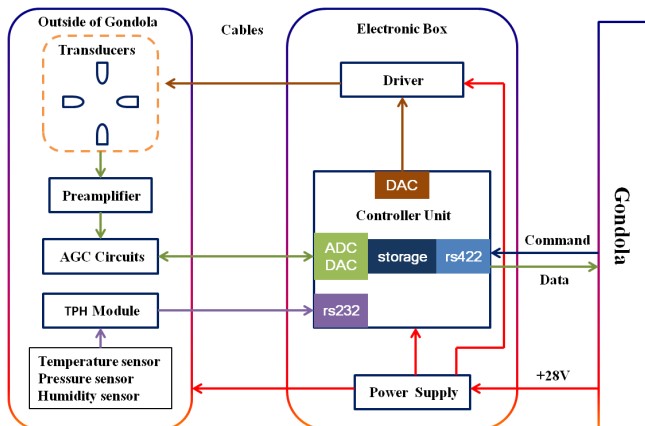

Figure 3 Block diagram of the electrical system associated with the acoustic anemometer.




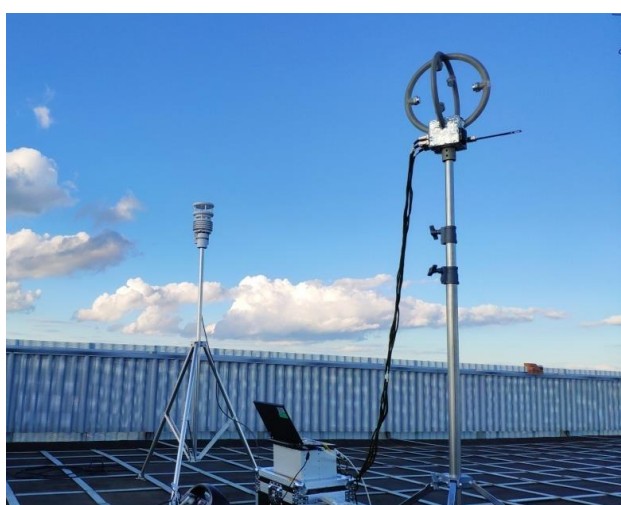

Figure 4 Photograph of the acoustic anemometer developed in this study alongside a commercial anemometer at the summit of the Wuling Mountain.


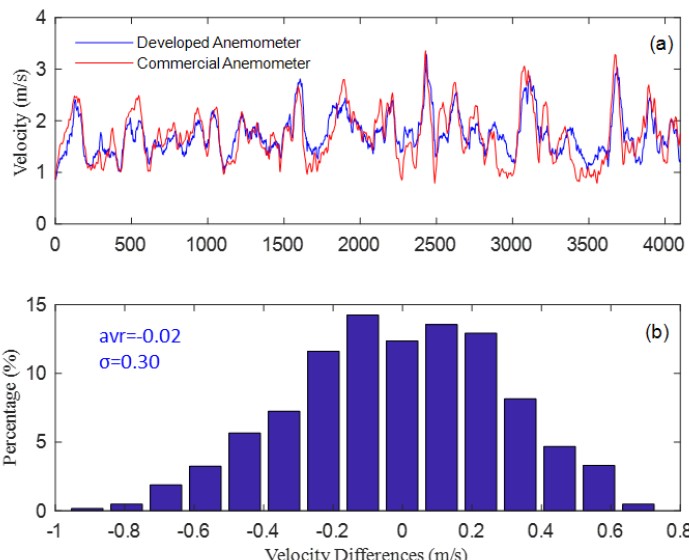

Figure 5 Comparison of observations at the top of Wuling Mountain between the anemometer developed in this study and the commercial anemometer. In (a), the blue line represents data obtained from the anemometer developed in this study while the red line represents data obtained from the commercial anemometer. In (b), the distribution of the differences between the commercial anemometer and our anemometer are shown. All data in this figure are averaged in 1-minute intervals to minimize error caused by the different sampling times of the two anemometers.




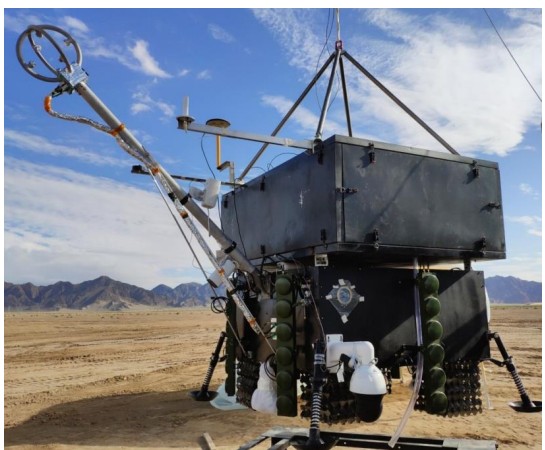


Figure 6 Photograph of the gondola with payloads prior to ascent on 4 September 2019.





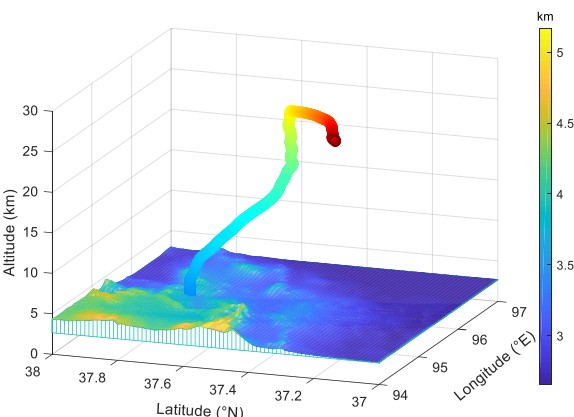

Figure 7 Flight trajectory (color line, the dark red stands for the end of the float flight) of the gondola during the 4 September 2019 experiment. The colorbar represents the topography of the experiment site.



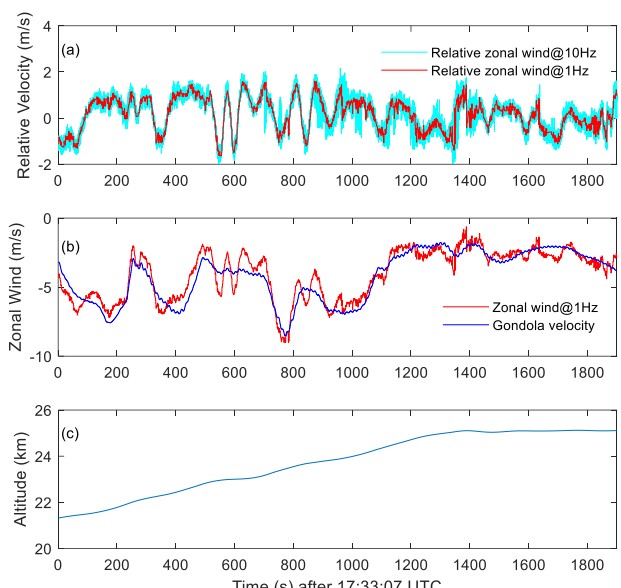


Figure 8 For a 1900-s period starting at 17:33:07 UTC: (a) relative zonal wind speed measured by the anemometer at update rates of 10Hz (cyan) and 1Hz (red), respectively, (b) comparison between zonal wind speed (red) and gondola zonal movement speed (blue), and (c) gondola altitude.






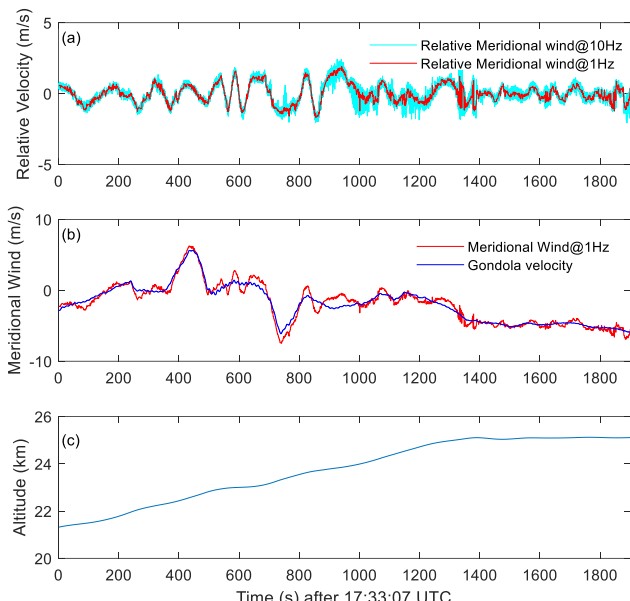

Figure 9 As in Fig.8, but for the meridional wind.





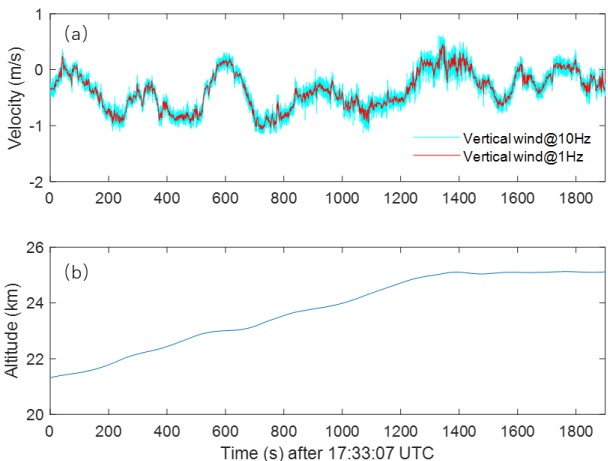


Figure 10 (a) vertical wind speed measured by the anemometer at update rates of 10Hz (cyan) and 1Hz (red), respectively, and (b) gondola altitude.





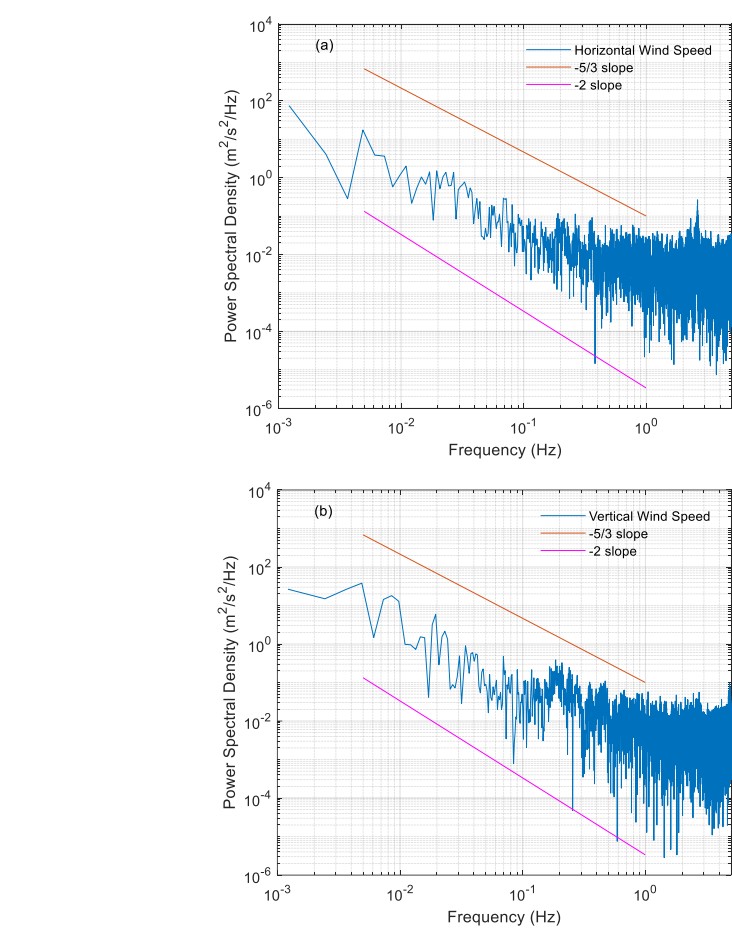


Figure 11 Spectral analysis of the 10Hz measurements during the float flight period from 1500s to 1900s
starting at 17:33:07 UTC: (a) relative horizontal wind speed, and (b) vertical speed. The orange line and
magenta line in each case indicates the theoretical spectral trend of -5/3 and -2, respectively.





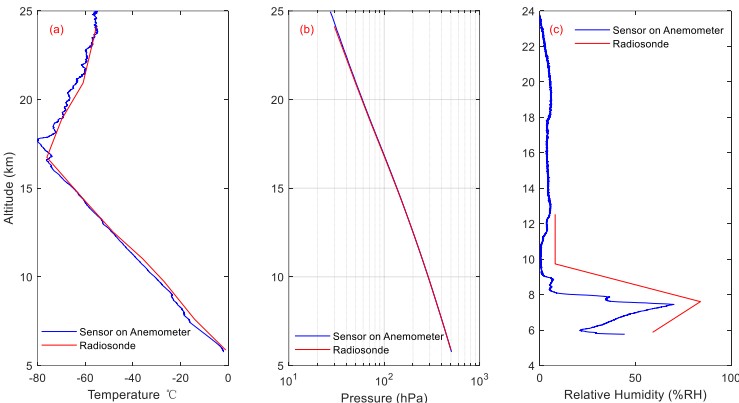

Figure 12 Comparisons of (a) temperature, (b) pressure, and (c) relative humidity between measurements
from sensors on the anemometer and radiosonde data obtained from the Golmud Observation Station
approximately four hours earlier.