# Peer review of "Development of an *in situ* Acoustic Anemometer to Measure Wind in the Stratosphere for SENSOR"

_Atmospheric Measurement Techniques, 2021_

## Author Comment (AC1)

**Responses from authors to Reviewer 1's comments**

**Ref. No.: amt-2021-424**

**Title: Development of an in situ Acoustic Anemometer to Measure Wind in the Stratosphere for SENSOR**

Liang Song[1,2], Xiong Hu[1], Feng Wei[1], Zhaoai Yan[1,3], Qingchen Xu[1], Cui Tu[1,3]

[1]Key Laboratory of Science and Technology on Environmental Space Situation Awareness, National Space Science Center, Chinese Academy of Sciences, Beijing 100190, China

[2]College of Earth Sciences, University of Chinese Academy of Sciences, Beijing 100049, China

[3]College of Materials Science and Opto-Electronic Technology, University of Chinese Academy of Sciences, Beijing 100049, China

*Reviewer 1's comments:*

*This manuscript focuses on the description of the development of a sonic anemometer designed to perform measurements in the stratosphere (with a sampling rate of 10Hz) on board of high altitude research balloons. A claim is made in the abstract that "Developing this anemometer was necessary, as there is no existing commercial off-the-shelf product, to the authors' knowledge, capable of obtaining in situ wind measurements on a high-altitude balloon or other similar floating platform in the stratosphere". Clearly, the latter statement appears to be not accurate, as later on in the text the authors cite an article by Maruca et al. (appeared in AMT in 2017 : https://amt.copernicus.org/articles/10/1595/2017/) describing an experiment in which an off-the-shelf anemometer with minimal modifications was employed to perform three-dimensional velocity measurements of the wind in the stratosphere, with a sampling rate of 200Hz. The run performed by Maruca et al. produced data used to conduct a spectral analysis of the stratospheric wind, presented in the same AMT article. Previously, Banfield et al. developed and tested a homemade acoustic anemometer which operated up to an altitude of 33km, returning as well high resolution wind velocity measurements. In my opinion, the outcome of these 2016 and 2017 articles allows to say that operating a sonic anemometer in the stratosphere is by itself no news, which is the major problem I have with the present manuscript.*

*Indeed, main conclusions here are that the acoustic anemometer developed by the authors "obtained continuous wind velocity data at the floating altitudes of 24-25km" and "...preliminary spectral analysis demonstrate that the acoustic anemometer employed in this study can sense rapid changes in wind and is useful for researching small-scale wind fluctuations in the stratosphere", indeed similarly to what was done by Maruca et al. in 2017.*

*I consider very valuable the efforts made by the authors to develop a new acoustic instrument able to perform velocity wind measurements in the stratosphere, I really think this is needed and I strongly encourage them to pursue with further developments of their instrument. However, in order for a probe to be worthy of becoming the subject of a scientific article, such instrument should either make it possible sets of observations which were not possible in before, or the measurements collected in runs of the newly developed probe must be used to produce original analyses and results. The latter should address one or more science cases that need do be described and thoroughly discussed in the draft proposed for publication. For these reasons, I cannot suggest the present manuscript for the publication in AMT.*

**Answers to the Reviewer1's comments:**

**Thank you very much for your time and efforts reviewing this study. The answers that we have made based on the reviewer's comments are discussed below (the comments are shown in italics and blue while responses in non-italics and red).**

*This manuscript focuses on the description of the development of a sonic anemometer designed to perform measurements in the stratosphere (with a sampling rate of 10Hz) on board of high altitude research balloons. A claim is made in the abstract that "Developing this anemometer was necessary, as there is no existing commercial off-the-shelf product, to the authors' knowledge, capable of obtaining in situ wind measurements on a high-altitude balloon or other similar floating platform in the stratosphere". Clearly, the latter statement appears to be not accurate, as later on in the text the authors cite an article by Maruca et al. (appeared in AMT in 2017 : https://amt.copernicus.org/articles/10/1595/2017/) describing an experiment in which an off-the-shelf anemometer with minimal modifications was employed to perform three-dimensional velocity*

*measurements of the wind in the stratosphere, with a sampling rate of 200Hz. The run performed by Maruca et al. produced data used to conduct a spectral analysis of the stratospheric wind, presented in the same AMT article. Previously, Banfield et al. developed and tested a homemade acoustic anemometer which operated up to an altitude of 33km, returning as well high resolution wind velocity measurements. In my opinion, the outcome of these 2016 and 2017 articles allows to say that operating a sonic anemometer in the stratosphere is by itself no news, which is the major problem I have with the present manuscript.*

**Response-1:**

Thank you very much for your suggestions. The claim here is indeed inappropriate. What we want to clarify is that no off-the-shelf equipment can be used directly on our balloon, which is flying at an altitude of about 25km. We will modify the claim according to your suggestions as follows.

"Developing this anemometer was necessary, as there is no existing commercial off-the-shelf product, to the authors' knowledge, capable of obtaining in situ wind measurements on a high-altitude balloon we used in SENSOR campaign, which is floating at an altitude of about 25km".

*Indeed, main conclusions here are that the acoustic anemometer developed by the authors "obtained continuous wind velocity data at the floating altitudes of 24-25km" and "...preliminary spectral analysis demonstrate that the acoustic anemometer employed in this study can sense rapid changes in wind and is useful for researching small-scale wind fluctuations in the stratosphere", indeed similarly to what was done by Maruca et al. in 2017.*

**Response-2:**

Thank you for your comment. To ensure that the anemometer can function properly at the floating altitude (~25km) of our balloon, we had taken further improvements based on drawing experiences from their work. Different from theirs, we choosed transducers at lower resonant frequency according to the analysis of acoustic signal propagation attenuation in the atmosphere, and designed an Automatic Gain Control (AGC) circuit to adjust received signal gain levels with altitude range. As a result of these efforts, we obtained measurements during float flight. For data analysis, Maruca et al. (2017) presented spectra of data during ascent, which were in the Euler frame of reference. And what we had done was to evaluate the Lagrangian spectrum slope with frequency in the inertial subrange using measurements during float flight of a high-alitude balloon. These aspects are different from what was done by Maruca et al. in 2017, and are also the major contributions of our work.

*I consider very valuable the efforts made by the authors to develop a new acoustic instrument able to perform velocity wind measurements in the stratosphere, I really think this is needed and I strongly encourage them to pursue with further developments of their instrument. However, in order for a probe to be worthy of becoming the subject of a scientific article, such instrument should either make it possible sets of observations which were not possible in before, or the measurements collected in runs of the newly developed probe must be used to produce original analyses and results. The latter should address one or more science cases that need do be described and thoroughly discussed in the draft proposed for publication. For these reasons, I cannot suggest the present manuscript for the publication in AMT.*

**Response-3:**

Thank you very much for your appreciation and encouragement of our work. Our article mainly focused on the development of an acoustic anemometer for the SENSOR campaign and the demonstration of measurements obtained from the flight experiment. As discussed in Response-2, to our knowledge, the efforts we had taken to accommodate our acoustic anemometer to the high-altitude atmosphere and the analysis of the Lagrangian spectra in the inertial subrange using measurements during float flight of a high-alitude balloon have not been reported before.

We have expanded and re-combed the part of spectral analysis in the revised manuscript. We appreciate your suggestions to show more science cases in the manuscript. These work are indeed in process and will be presented in another article because they are considered to deviate from the topic of this paper.

---

## Author Comment (AC2)

**Responses from authors to Reviewer 2's comments**

**Ref. No.: amt-2021-424**
**Title: Development of an in situ Acoustic Anemometer to Measure Wind in the Stratosphere for SENSOR**

Liang Song[1,2], Xiong Hu[1], Feng Wei[1], Zhaoai Yan[1,3], Qingchen Xu[1], Cui Tu[1,3]
[1]Key Laboratory of Science and Technology on Environmental Space Situation Awareness, National Space Science Center, Chinese Academy of Sciences, Beijing 100190, China
[2]College of Earth Sciences, University of Chinese Academy of Sciences, Beijing 100049, China
[3]College of Materials Science and Opto-Electronic Technology, University of Chinese Academy of Sciences, Beijing 100049, China

*Reviewer 2's comments:*

*The authors introduce a self-made ultrasonic anemometer (hereafter called "sonics") for wind velocity measurements based on a drifting stratospheric balloon in an altitude of about 25 km.*

*It is claimed that these are the first measurements with high resolution at those altitudes because off-the-shelf sonics do not reliable run in those extreme environments due to poor signal-to-noise ratio.*

*The authors claim that the main motivation for their work is "To the authors' knowledge, there is no existing commercial off-the-shelf product that can obtain in situ wind measurements at altitudes exceeding 20 km" which is then also immediately refuted to a large extent in a cited article by Marcua et al. which presents sonic data observed in an altitude of 18 km which is not much less than the mentioned 20 km.*

*The authors start their sensor evaluation with a ground-based intercomparison with a commercial sonic with much less temporal resolution which has been designed for weather observations and not research. There is no doubt about the question if the new developed sensor works in general and provides reasonable data, however, this simple experiment does not provide high quality data needed for a detailed characterization of the sensor performance. I got no idea about the absolute accuracy or even the sensor resolution of the new sonic based on carefully performed observations.*

*The technical description of the new sonic is at many places too detailed compared to the real important things. For example, the discussion about the transducer separation L is vague; so why did you choose 20 cm which is longer than for most commercial devices and how do you conclude that this compromise is working? There are many other places which provides information which does not really help to better understand your device. That a "controller board serves as a brain" does not provide any useful information. Phrases as "extreme environments" do not help; please provide numbers and I suggest avoiding such vague phrases and technical details which are not necessary to understand the real important things. That a circuit is protected by a fuse is standard and not worth to be mentioned here but instead I would like to know in which way you have overcome technical limitations you mentioned at the beginning and why is your system running in 25 km height and others not (if true).*

*About the data analysis:*

*You mentioned spikes in the observations as technical problems but no details are given how they have been removed. Also*

*possible reasons for that spikes are only very briefly mentioned but if this a problem for high altitude measurements with sonics than a more detailed discussion would be interesting.*

*Measuring vertical wind speeds from a moving platform is extremely challenging and strongly depends on the accuracy of the measured pitch angle; however, there is absolutely no discussion about this issue.*

*The section about the power spectral analysis is very brief and vague but you draw the simple conclusion that the newly developed sonic works well - this is too less for a scientific analysis and does reads more like a short progress report. To be convinced that this sensor provides useful data for scientific analysis - in particular for turbulence analysis - much more work is needed. From my point of view there is no proof that this ultrasonic works better in the stratosphere than other sensors. I had a brief look into the corresponding section of Marcua et al.:   their sonic provides two decades higher spectral resolution which is striking. The only differences which might justify a new publication is that Marcua et al provide observations up to "only"18 km and your data has been observed at 25 km - however, with less resolution.*

*Although I greatly appreciate the development of the new sonic, the manuscript in its current state does not warrant publication. The progress compared to other similar systems is not convincingly presented and the analysis methods of the acquired data - both on the ground as a comparison and on the balloon - are rather simplistic.*

**Answers to the Reviewer2's comments:**
**Thank you very much for your time and efforts reviewing this study. The answers that we have made based on the reviewer's comments are discussed below (the comments are shown in italics and blue while responses in non-italics and red).**

*The authors introduce a self-made ultrasonic anemometer (hereafter called "sonics") for wind velocity measurements based on a drifting stratospheric balloon in an altitude of about 25 km.*
*It is claimed that these are the first measurements with high resolution at those altitudes because off-the-shelf sonics do not reliable run in those extreme environments due to poor signal-to-noise ratio.*
*The authors claim that the main motivation for their work is "To the authors' knowledge, there is no existing commercial off-the-shelf product that can obtain in situ wind measurements at altitudes exceeding 20 km" which is then also immediately refuted to a large extent in a cited article by Marcua et al. which presents sonic data observed in an altitude of 18 km which is not much less than the mentioned 20 km.*

**Response-1:**
Thank you for your comments. The claim here is indeed inappropriate. What we want to clarify is that no off-the-shelf equipment can be used directly on our balloon, which is flying at an altitude of about 25km. We will modify the claim in the revised manuscript as follows.

"Developing this anemometer was necessary, as there is no existing commercial off-the-shelf product, to the authors' knowledge, capable of obtaining in situ wind measurements on a high-altitude balloon we used in SENSOR campaign, which is floating at an altitude of about 25km".

*The authors start their sensor evaluation with a ground-based intercomparison with a commercial sonic with much less temporal resolution which has been designed for weather observations and not research. There is no doubt about the question if the new developed sensor works in general and provides reasonable data, however, this simple experiment does not provide high quality data needed for a detailed characterization of the sensor performance. I got no idea about the absolute accuracy*

*or even the sensor resolution of the new sonic based on carefully performed observations.*

**Response-2:**

The purpose of intercomparison experiment is to verify the reliability of our device. As there is no available wind tunnel, we choose to carry out the comparative test with a commercial anemometer on the top of the mountain. In order to minimize the difference caused by the measurement time and location during the comparison of the two devices, we compared the average values over a period of time, so we didn't care about the resolution of the reference device. In the revised manuscript we have added the discussion of the resolution of our anemometer.

*The technical description of the new sonic is at many places too detailed compared to the real important things. For example, the discussion about the transducer separation L is vague; so why did you choose 20 cm which is longer than for most commercial devices and how do you conclude that this compromise is working? There are many other places which provides information which does not really help to better understand your device. That a "controller board serves as a brain" does not provide any useful information. Phrases as "extreme environments" do not help; please provide numbers and I suggest avoiding such vague phrases and technical details which are not necessary to understand the real important things. That a circuit is protected by a fuse is standard and not worth to be mentioned here but instead I would like to know in which way you have overcome technical limitations you mentioned at the beginning and why is your system running in 25 km height and others not (if true).*

**Response-3:**

We greatly appreciate for your suggestions and the technical details have been revised in the manuscript.

For choosing transducer separation L, the shadowing effects of the sensor and the absorption attenuation of acoustic signals are mainly considered. In order to minimize the influence of transducer shadowing effects, the distance between sensors should be as far as possible, but this will reduce the signal-to-noise ratio of received signals. Figure 1 shows the absorption attenuation with different transducer separation at the altitude of 25km. It can be seen that the attenuation at 20cm is about 1dB more than that at 10cm, but we transmit 40kHz signal, which is lower than that of most commercial devices ( as far as we know, most commercial acoustic anemometers operate at frequencies above 100 kHz). As discussed in section 2.2.1 of the manuscript, the received signal intensity with frequency of 40kHz is at least 10dB higher than that of signals with frequencies of above 100kHz. Thus our anemometer can obtain higher SNR than most commercial ones, though L is longer than theirs. Also, it is very important that we have tested the sensors with distance of 20cm in a vacuum and verified that received signals can be well distinguished from noises.

[Figure]

Figure 1 The absorption attenuation of acoustic signals with different transducer separation at the altitude of 25km

we have deleted the relevant technical details which are not essential to the contents of the paper and clarified the vague phrases. We did specify the extreme environments in the introduction part of the manuscript, but we ignored to mention this here. Thank you for for finding the oversight and we have replaced the term by "extreme enviroments with low temperatures of ~70°C and pressures of about 30hPa".

As discussed in section 2.2.1 of the manuscript, low atmospheric pressure is the main challenge faced by acoustic anemometers, since acoustic signals absorption attenuation in the atmosphere will be increased with height. All the efforts we made are to obtain the received signal to noise ratio (SNR) as high as possible, so we have taken improvement measures such as reducing the frequency of acoustic signals and designing an Automatic Gain Control (AGC) circuit. Those are different to other acoustic anemometers and can ensure the anemometer accommodate to the high-altitude atmosphere. About reducing the frequency of acoustic signals, we have choosed sensors with resonant frequency of 40kHz, the received signal intensity is at least 10dB higher than that of signals with frequencies of above 100kHz, although the transducer separation is longer than for most commercial devices, which just causes about 1dB more signal attenuation (as shown in Figure 1). The AGC circuit has been used to adjust its gain levels by altitude range to amplify received signals attenuated with height. And our anemometer has been tested in a thermal vacuum that is used to simulate the atmospheric environment at the altitude of 25km and the result shows that our device operate properly.

*About the data analysis:*

*You mentioned spikes in the observations as technical problems but no details are given how they have been removed. Also possible reasons for that spikes are only very briefly mentioned but if this a problem for high altitude measurements with sonics than a more detailed discussion would be interesting.*

**Response-4:**
Thank you for your suggestion. We have clarified the details in the manuscript. Spikes in the wind speed were caused by the large attenuation of acoustic signals under low pressures, which led to the misjudgement of one period (25 microseconds) or even several periods in the propagation time between one pair of transducers during cross-correlation processing. The time of one period will cause the wind speed in this direction to change by about 11m/s, which seems to be a spurious spike. This will

be eliminated by adding or substracting one period in the propagation time, depending on whether the spike is larger or lower than the front and back values.

*Measuring vertical wind speeds from a moving platform is extremely challenging and strongly depends on the accuracy of the measured pitch angle; however, there is absolutely no discussion about this issue.*

**Response-5:**
Thank you for your comments. The issue is very important and we have added the discussion about this in the revised manuscript. The attitude, include the angular rates and attitude angles, are continuously monitored by the platform with the Inertial Navigation Sensor (INS) installed on the gondola. These data are provided once a second and the accuracy of the measured pitch angle and roll angle is 0.1°. From these data, we can conclude that the pitch angle of the platform changes slowly and small during level flight. Then with the angle relationship between sensors and the gondola, vertical wind can be obtained by coordinate transformation.

*The section about the power spectral analysis is very brief and vague but you draw the simple conclusion that the newly developed sonic works well - this is too less for a scientific analysis and does reads more like a short progress report. To be convinced that this sensor provides useful data for scientific analysis - in particular for turbulence analysis - much more work is needed. From my point of view there is no proof that this ultrasonic works better in the stratosphere than other sensors. I had a brief look into the corresponding section of Marcua et al.: their sonic provides two decades higher spectral resolution which is striking. The only differences which might justify a new publication is that Marcua et al provide observations up to "only" 18 km and your data has been observed at 25 km - however, with less resolution.*

**Response-6:**
We have expanded and re-combed this part in the revised manuscript. About the resolution, the internal repetition rate of measurements we developed is 10Hz. In order to improve the received SNR, the original sampled signals within 1s are accumulated in data processing, so the data update rate we given by default is 1Hz and in spectral analysis we have used the 10Hz data. The resolution has already met our scientific requirements. This is indeed much lower than that of Maruca et al., which is one aspect of the device we are improving so that it can obtain fluctuations on a smaller temporal scale.

We think that the major contributions of our work are the efforts we have taken to accommodate our anemometer to the high-altitude atmosphere and the analysis of the Lagrangian spectra in the inertial subrange using measurements during float flight of a high-alitude balloon, though there are much work to be done in improving our device, we really hope to share these work with the scientific community.